# G Protein-Coupled Receptor Dimerization—What Next?

**DOI:** 10.3390/ijms25063089

**Published:** 2024-03-07

**Authors:** Marta Dziedzicka-Wasylewska, Agnieszka Polit, Ewa Błasiak, Agata Faron-Górecka

**Affiliations:** 1Department of Pharmacology, Maj Institute of Pharmacology, Polish Academy of Science, Smętna Street 12, 31-343 Kraków, Poland; wasyl@if-pan.krakow.pl; 2Department of Physical Biochemistry, Faculty of Biochemistry Biophysics and Biotechnology, Jagiellonian University, Gronostajowa 7, 30-387 Kraków, Poland; a.polit@uj.edu.pl (A.P.); ewa.blasiak@uj.edu.pl (E.B.)

**Keywords:** G protein-coupled receptor, heterodimers, GPCR signaling, lipid–membrane interactions, cholesterol, anionic lipids, polyunsaturated fatty acids

## Abstract

Numerous studies highlight the therapeutic potential of G protein-coupled receptor (GPCR) heterodimers, emphasizing their significance in various pathological contexts. Despite extensive basic research and promising outcomes in animal models, the translation of GPCR heterodimer-targeting drugs into clinical use remains limited. The complexities of in vivo conditions, particularly within thecomplex central nervous system, pose challenges in fully replicating physiological environments, hindering clinical success. This review discusses examples of the most studied heterodimers, their involvement in nervous system pathology, and the available data on their potential ligands. In addition, this review highlights the intricate interplay between lipids and GPCRs as a potential key factor in understanding the complexity of cell signaling. The multifaceted role of lipids in modulating the dynamics of GPCR dimerization is explored, shedding light on the elaborate molecular mechanisms governing these interactions.

## 1. Introduction to G-Protein Coupled Receptors

G protein-coupled receptors (GPCRs) act as receptors for many different signals and are important for the functioning of an organism; they are differentially expressed in many cell types in the body, including neurons. Thorough analyses of GPCR repertoires of human and mouse [1] have shown that over 90% of them are expressed in the brain. On the other hand, as the authors stated, “…the profiles of most GPCRs are unique, yielding thousands of tissues- and cell-specific receptor combinations for modulation of physiological processes”.

It has been estimated that ca. 30% of all currently marketed drugs are modulators of specific GPCRs, and these receptors are still within the area of interest of medicinal chemistry [2].

Mammalian GPCRs constitute a superfamily of membrane proteins, which are grouped into four classes: A, B, C and F/S [3]. Although quite diverse in many ways, all members of the GPCR superfamily have seven transmembrane domains. The pioneering work of Krzysztof Palczewski, concerning the elucidation of the crystal structure of rhodopsin [4], had a fundamental influence on research on GPCR functioning. Research on the structure and structural alterations of GPCRs upon ligand binding has been awarded the Nobel Prize in 2012.

The complex 3D structure of GPCRs is quite unique among receptors (except for ligand-gated ion channels), and still remains a challenge for the scientific community [5]. Most probably, such a complex structure provides the means for various interactions—not only with ligands but also with other proteins, including GPCRs themselves.

A commentary on GPCR heterodimerization studies and the conclusions drawn from these studies, as well as on the involvement of other components of the signal transduction machinery, such as G proteins or membrane lipids, is provided below. Most examples come from studies on the heterodimerization of adenosine A2A and dopamine D2 receptor, dopamine D1 and D2 receptors and opioid receptors, as these are the studies with the greatest potential for use in the clinic.

## 2. GPCR Heterodimerization—Historical Background

The first evidence of the interaction and mutual regulation between GPCRs emerged in the 1970s from biochemical studies, indicating the negative cooperativity between beta-adrenergic receptors [6], and from behavioral studies concerning adenosine and dopamine receptors [7].

The turn of the century has brought two fundamental direct facts, the first concerning GABA-B receptors [8] and the second one concerning opioid receptor kappa and delta [9].

GABA-B receptors are metabotropic receptors (different from GABA-A receptors, which are ligand-gated ion channels). It has been shown that they form obligatory heterodimers: one of the protomers is responsible for membrane localization and the binding of the ligand, and the other one for activating signal transduction. Each needs a partner and does not function without the other.

Significant results have been obtained for opioid receptors kappa and delta, which form heterodimers mostly in the spinal cord. A promising hypothesis was formed: if the compounds targeting these heterodimers were found, they would be extremely important as analgetic drugs, free of the serious side effects observed with morphine.

To make an order among a plethora of studies indicating the formation and importance of various heterodimers which have followed these important discoveries, the International Union of Basic and Clinical Pharmacology issued special recommendations for the recognition of GPCR hetero- or multimers [10]. At least two conditions from the list below are obligatory to consider interactions of a given pair of receptors as significant:Both protomers must reside in the same cell or its compartment;The phenomenon of physical interaction of a given pair of receptors must be demonstrated in native tissue, no matter what technique is used;The identification of unique pharmacological features of a given heteromer is important;Pharmacological (biochemical) response of a given heteromer must disappear when one of the protomers is absent (e.g., in transgenic animals following siRNA silencing).

Heterodimerization is important at various levels of receptor functioning: from biosynthesis and membrane localization to pharmacological diversity, signal transduction, and receptor internalization [11]. A lot of research has been dedicated to structural aspects of heterodimer formation. The involvement of individual transmembrane domains has been extensively studied using the model rhodopsin system, as well as various pairs of GPCRs; there is no general rule, and each pair has to be studied individually [12]. Crystal- or NMR-based 3D structure determination [13] and in silico approaches [14] may also provide useful clues about the contact interface, and help in designing ligands. The reported allosteric modulation taking place within heteromers suggests that this interface is dynamic and sensitive to small conformational changes [15,16], which may include a not-yet-recognized contribution of the lipid environment in shaping receptor conformation [17].

## 3. Heterodimerization in the Nervous System

The potential role of heterodimerization of GPCRs is especially important in the central nervous system (CNS); there is great diversity in the localization of a given receptor in the brain regions (and other organs), which is responsible for serious side effects upon pharmacotherapy. Physical interactions between receptors (hetero-dimerization) can take place only if they are localized in the same neuron; therefore, the pharmacological targeting of heterodimers localized in appropriate type of neurons should lead to the minimization of unwanted side effects of potential drugs. Various methodological approach to study GPCR co-expression and heterodimer formation has been recently described in detail [18]. As the authors state, “…The occurrence of GPCR homo- and heteromers in artificial systems is generally well accepted, but more specific methods are necessary to address GPCR oligomerization in the brain...” Indeed, in spite of the constantly developing research methodologies, estimating in vivo the percentage of neurons co-expressing a given receptor pair, not to mention the density of heterodimers they can form, is not an easy task. We still encounter problems that are not even mentioned in numerous studies.

A good example is research on the co-localization of dopamine D1 and D2 receptors in the brain—these receptors are expressed in medium spiny neurons (MSN), which represent up to 95% of the neuronal population in the striatum [19]. All MSNs use gamma-aminobutyric acid as a neurotransmitter, but they can be divided into two subpopulations: ca. half of the MSNs express dopamine D1 receptor and as co-transmitters. They contain substance P and dynorphin, and project mainly to the substantia nigra pars reticulata and the entopeduncular nucleus. The other half of MSNs express dopamine D2 receptor and contain enkephalin as co-transmitter; they innervate mainly pallidum [20]. For a long time, the concept of two subpopulations of MSNs had its place in understanding the functioning of the basal ganglia, although synergism of the action between the ligands of both receptors was noted in many experimental situations, and attempts to explain the molecular basis of this phenomenon varied [21].

The demonstration that D1 and D2 receptors can reside on the same neuron has become important not only because of the fashionable phenomenon of GPCR heterodimerization, but also because of the possibility of explaining the synergistic action of the two receptors’ ligands. However, it was not easy to demonstrate reliably the size of the MSN subpopulation expressing both dopamine receptors. In fact, depending on the methodology used, various percentages of such neurons have been reported, ranging from nearly 100% to very low [22]. Such differences are usually explained by the variety of methods and experimental animals explored. Using newer technology (e.g., transgenic mice expressing fluorescent reporters for dopamine D1 and D2 receptors), a relatively low number of MSNs expressing both dopamine receptors has been reported [22]: in the mouse dorsal striatum ca. 2% of such neurons were identified (whereas it should also be remembered that a similarly low percentage of cholinergic interneurons in the striatum have very important functions), and more of them in the nucleus accumbens (ca. 15% in the shell and ca. 7% in the core). The proportion of these receptors co-expressed in the same neuronal cell which are able to form heterodimers—constitutively or upon stimulation—has not been estimated so far.

An interesting example that the co-expression of two receptors in the same neuron does not necessarily imply heterodimer formation was recently published by Arttamangkul et al. [23]. Their results indicate that while mu and delta opioid receptors are expressed in single striatal cholinergic interneurons, the two receptors function independently in live brain slices.

## 4. Dopamine D1 and D2 Receptor Heterodimers

It has been widely accepted that well-defined dopaminergic pathways are responsible for different functions. Generally, it has been shown that nigrostriatal pathway governs motor functions, and it is dysregulated in Parkinson disease due to the loss of dopamine-producing cells in the substantia nigra system. The mesolimbic dopaminergic system is important for regulating rewarding and is most probably dysregulated in drug addiction. Mesocortical and mesolimbic DA systems are involved in the pathomechanisms underlying schizophrenia and psychoses. In the tuberoinfundibular system (linking the hypothalamus and pituitary gland), dopamine plays the role of a hormone, regulating the secretion of prolactin. All those systems are functioning via dopamine receptors, so if a blockade by antipsychotic drugs of the D2 receptor is necessary, there is a blockade of the same receptor located in the pituitary gland at the same time, and prolactin secretion will be disinhibited. Similarly, the D2 receptor in the nigrostriatal pathway would be blocked, leading to parkinsonian-like symptoms [24]. Hence, the idea that there is a portion of neurons expressing both dopamine receptors, D1 and D2 [25], and that they can form heterodimers was promising in the context of the search for compounds able to act via D1–D2 heterodimers, being then more selective and causing less side effects. Although promising, this idea was controversial for many years [26], but currently, due to extensive work [27,28], the concept has been proven. Moreover, in vitro studies demonstrated that dopamine receptors can form heterodimers, especially upon the concomitant stimulation of both D1 and D2 receptors, implicating that more heterodimers are encountered under the conditions of increased dopaminergic transmission [29,30,31]. Additional experiments indicated that D1–D2 receptor heterodimers activate different intracellular signals: the heterodimer stimulates phospholipase C via Gq G protein, leading to an increase in intracellular calcium concentration [32]. These results opened up a few questions concerning schizophrenia. Since this complex disease is treated with dopamine receptor antagonists, the main hypotheses concerning the biochemical mechanism underlying this disease still take into account overactivity of dopamine transmission. This means that dopamine receptors are overstimulated. Interestingly, an increase in D1–D2 heterodimers has been shown in the brains of patients suffering from schizophrenia (post-mortem studies) [33]. Additionally, an increase in calcium-binding proteins, calcyon and NCS-1, has also been reported [34,35]. There are many antipsychotic drugs available, but one of them, clozapine (CLZ), has a unique mode of action. It is effective for both negative and positive symptoms of schizophrenia, but the precise mechanism of its action is not fully clear. It also has many serious side effects, the most important being agranulocytosis, myocarditis, cardiovascular, and respiratory effects [36]. Therefore, there are continuous studies that aim to find the actual molecular mechanism of action of CLZ in order to mimic it with a new compound, possibly devoid of side effects but equally potent in treating the symptoms. In vitro studies, using an advanced fluorescence approach, have shown that CLZ is able to uncouple heterodimers of D1–D2 receptors [30]. Uncoupling of the D1–D2 heterodimer by CLZ might therefore be an important mechanism leading to the normalization of over-active dopamine transmission in schizophrenia.

An increase in the D1–D2 heterodimer has also been shown in the brains of patients suffering from another complex neuropsychiatric disease, major depression [37]. Additionally, the domains responsible for the physical interaction of these two receptors have been identified, and the peptide able to uncouple the heterodimer has been synthesized. In animal studies, it has been shown that this peptide acts similarly to a well-known antidepressant drug, imipramine, in behavioral tests used commonly to estimate the potency of antidepressant compounds.

A separate group of studies concerns the effects of THC on D1–D2 heterocomplexes. As has been postulated, the D1–D2 heterodimer contributes to the aversive-, anhedonic-, depressive-, anxiogenic-like, and diminished motivational behaviors in rodents. Studies with the use of adult nonhuman primates indicate, that repeated tetrahydrocannabinol (THC) administration upregulated D1–D2 heterodimers [38,39,40,41], accompanied by the modulation of specific proteins involved in the D1–D2 heterodimer signaling pathway, including increased calcium signaling markers and decreased cAMP (cyclic adenosine monophosphate)-linked signaling proteins [42]. These changes have been interpreted as contributing to reduced reward sensitivity, negative emotionality, and other behavioral impairments associated with cannabis use.

These studies were followed by investigating the effect of chronic THC administration and a withdrawal phase on the D1–D2 heterodimer expression and function [43]. It has been shown that chronic THC increased D1–D2 heterodimer expression and function in the nucleus accumbens; the effect was associated with an increased dynorphin expression and kappa opioid receptor activation. The authors interpreted these results as leading to a reduction in dopamine release, possibly triggering anxiogenic- and anhedonic-like behaviors after daily THC administration that persist for at least 7 days after drug cessation. These findings introduce a therapeutic strategy to alleviate the negative symptoms associated with cannabis use and withdrawal.

## 5. GPCR Heterodimer-Induced Intracellular Signaling

The concept of the existence of heterodimers of different receptors is very promising, but in general, most studies are limited to determining in which experimental or pathological situations the level of the respective heterodimers is modulated. But what are the actual consequences for the functioning of a cell with an increased (or decreased) number of heterodimers? The answers to such questions have been sought for years by Dr S. George’s group, reviewed recently by Misganaw [44]. Within the dopamine D1–D2 receptor heterodimer, they have demonstrated the altered signal that is received by a cell on which such a heterodimer exists and is activated. Namely, D1–D2 receptor heterodimers were found to signal through the activation of Gαq/11 (in contrast to “canonical” pathways, i.e., D1 and D2 receptor-mediated signaling via the established Gαs/olf and Gαi/o, respectively), leading to intracellular calcium release, phosphorylation of calcium calmodulin kinase II (CAMKII), and increased brain-derived neurotrophic factor (BDNF) expression in the nucleus accumbens (NAc) and the ventral tegmental area (VTA) [32,33]. The same group has shown that D1–D2 receptor heterodimer signaling increased the expression of glutamic acid decarboxylase 67 (GAD67), the major enzyme in GABA synthesis [45]. However well-documented these data may be, it should be remembered that the SKF83595 compound was used as a specific heterodimer agonist (Figure 1A), which is criticized by other research groups, since its significant cross-reactivity to other receptors has been shown. It has also been suggested that the mechanism of D1–D2 receptor–mediated calcium signaling may largely involve downstream signaling pathways and may not be completely heteromer-specific [46,47,48].

Apart from the data obtained with the SKF83595 ligand, which are not accepted by everyone, an increase in D1–D2 receptor heterodimers has been demonstrated in the brains of patients suffering from depression [37] and schizophrenia [49], and also in animal models following the administration of cocaine [39] or tetrahydrocannabinol (THC) [42]. An especially interesting finding of that last study is that cannabidiol attenuated most of THC-induced neuroadaptations. Further analyses have shown that chronic THC increased NAc D1–D2 receptor heterodimer expression and function, which resulted in increased dynorphin expression and kappa opioid receptor activation [43].

The above-mentioned studies indicate that an increase in the number of D1–D2 heterodimers has, in principle, an unfavorable effect, including correlating with the occurrence of depression, so an attempt was made to uncouple the monomers using specially designed peptides. And indeed, an i.c.v. administration of such a peptide showed antidepressant effects in animal models [37,38]. This reveals the contribution of the D1–D2 receptor heterodimer in behavioral despair and may garner therapeutic benefits in disorders like depression and anxiety.

These studies indicated the possibility of a new approach to the treatment of depression, but to date, there is no indication of any preparation of this kind being available at clinics. In 2014, an interesting study was published on the intranasal delivery of the interfering peptide designed to disrupt the interaction between the D1 and D2 dopamine receptors [37]. Indeed, the delivery method was verified, and the significant antidepressant effects comparable to that of imipramine in behavioral tests have been shown [50]. Furthermore, it has been shown that the interfering peptide disrupted the D1–D2 interaction and it was detected in the prefrontal cortex after intranasal administration. As the authors concluded, “…This study provides strong preclinical support for intranasal administration of the D1–D2 interfering peptide as a new treatment option for patients suffering from major depressive disorders”. However interesting these results might be, nothing further is known about this peptide or similar ones in clinical trials.

## 6. Adenosine A2A–Dopamine D2 Heterodimer

The most prominent and highly exploited example of clinically important heterodimers is the one formed by adenosine A2A and dopamine D2 receptors. The A2A–D2 heterodimer is of especially high interest in view of its relevance for Parkinson’s disease, (resulting from a significant decrease in nigrostriatal dopamine neurotransmission). The allosteric properties of the heterodimer have been shown [51], which is the base for the use of an A2A receptor antagonist to increase the affinity of dopamine D2 receptor for dopamine. This is why A2A antagonists were a fantastic plan for compensatory mechanism of the non-dopaminergic approach for the therapy of Parkinson disease. It allowed for the idea of delaying the use of L-DOPA, which, over time stops working in patients. It is impossible to cite all or even the most important publications on the effects of A2A receptor antagonists in in vitro and in silico, pre-clinical, and clinical studies (interested readers can find more details in the work of [52]). Of the extensive and long-standing search for adenosine A2A receptor ligands, only one compound has been approved by the FDA for clinical use (Figure 1B). Istradefylline, (commercialized as Nourianz^®^) is a heterodimer-selective ligand which has been approved by the FDA for use as adjunctive treatment to L-DOPA in patients with Parkinson disease experiencing “off” episodes [53].

## 7. Opioid Receptor Heterodimers

Addressing the opioid receptor heterodimers has important place in the area of searching for effective antinociceptive drugs. Especially in the face of the “opioid crisis” (opioid misuse and abuse have become a major health concern worldwide), there is an urgent need to find effective therapeutics. Targeting heterodimers of opioid receptors seems to be an interesting strategy. The participation in the reduction in the nociceptive signal in chronic pain conditions has been clearly shown for four heteromers: three involve the δ receptor of which the expression is increased in neuropathic or inflammatory conditions and the selective targeting alleviates mechanical allodynia [54,55], δ-κ opioid, δ-CB1, and μ-δ. The fourth heteromer involves an association with μ and the chemokine receptor CCR5 [56].

In this area, however, a lot of research focuses on the synthesis of bivalent ligands, which consist of two ligands, each selective for one receptor type and linked together by a spacer of defined length. Hence, they are not so much specific for a given heterodimer, but simply target two receptors simultaneously, and since these compounds are linked, it can be assumed that they target two receptors which are close enough to each other to form a heterodimer. This area of research within the scope of opioid receptors has been thoroughly reviewed by Gaborit and Massotte [57], and Gunther et al. [58].

Generally, bivalent ligands share a limited capacity to cross the blood–brain barrier. This probably explains the absence of the rewarding effects when administered systemically, which can be viewed as an advantage over small opioid molecules. However, restricted targeting to peripheral receptor pairs may not provide maximal anti-nociception and limited central bioavailability hampers modulation of the central sensory mechanisms involved in emotional and cognitive aspects, thereby limiting their potential in clinical settings.

In contrast, small molecules efficiently cross the blood–brain barrier. Among them, bifunctional or mixed ligands were designed that simultaneously target two receptors [59]. Some of them, e.g., cebranopadol, a mixed non-selective opioid/NOP ligand, are in Phase III clinical trials for the treatment of severe chronic pain [60]. Attempts were thus made to develop small molecules that would selectively target heteromers. CYM51010 and eluxadoline were recently approved by the FDA, but for the treatment of irritable bowel syndrome [61].

To date, low-molecular-weight compounds targeting heteromers retain affinity and activity for receptor monomers at least to some extent. An interesting compound, synthesized as early as 2005, is 6-guanidinonaltrindole (6′GNTI) [62], a δ or κ receptor agonist (Figure 1C), raised hope for strong antinociceptive effectiveness devoid of side effects, especially since the reported low levels of δ-κ heteromers in the brain [63], but other studies have indicated that the selectivity of 6′GNTI toward δ-κ heteromers is only partial as this ligand is also a potent κ receptor agonist [64] and can activate μ-κ heteromers, although to a lesser extent.

Improving the selectivity of low-molecular-weight compounds constitutes a major pharmacological challenge to overcome.

## 8. Impact of Plasma Membrane on GPCR Functioning

An interesting concept has recently emerged concerning the perception of GPCRs, but also their heterodimers, in a broader context. Namely, the existence of GPCR-effector macromolecular membrane assemblies (GEMMAs) has been proposed, comprising of specific GPCR combinations, G proteins, effectors, and other associated plasma membrane-localized proteins [65]. The authors underline the importance of three primary components of membrane-delimited GPCR-mediated signaling pathways (GPCRs, G proteins, and plasma membrane effectors), and point to additional components that modulate or scaffold these core components (G protein-coupled receptor kinases and arrestins). GEMMAs are postulated as unique drug targets since they harbor distinct functional and pharmacological characteristics.

This publication is very significant, as few authors in the field of research on GPCRs and their heterodimerization have pointed out that these receptors do not act alone; they are present in the cell membrane in the company of many other proteins with which they interact. Ferre et al. [65] focused on describing these protein–protein interactions in depth within the context of the GEMMA concept. On the other hand, in the present work, we wish to highlight the impact of the membrane itself on GPCR function.

As integral components of the lipid membrane, GPCRs undergo diverse influences originating from both the membrane structure and the lipids themselves. The intricate nature of lipid membranes is further complemented by the asymmetry of the two lipid leaflets concerning their chemical composition and geometry. Consequently, the relationship between GPCRs and the membrane is complex at multiple levels. The membrane, responsible for the spatial organization of membrane proteins, establishes an environment with distinct properties such as lateral pressure of the surrounding lipid bilayer, membrane thickness, curvature, fluidity, and hydrophobic mismatch [66,67]. Simultaneously, the chemical heterogeneity of the cellular membrane lipids results in a broad range of specific interactions with GPCRs, reflecting the lipid’s chemical nature. These interactions, in conjunction with membrane effects, directly impact membrane-embedded GPCRs. They modulate the conformation of GPCRs and the potential interaction surface of the receptors, thereby influencing their function and physicochemical properties, including ligand binding affinity and oligomerization.

GPCRs display a unique lipid–protein interaction profile influenced not only by the GPCR itself, but also by the specific conformation of the receptor. These bound lipids not only modify the receptor’s surface, but can also act as allosteric modulators of receptor conformation [68]. The molecular signatures of receptor–lipid interactions and their functions for the majority of GPCRs, including dopamine receptors, are not yet fully understood. Nevertheless, they are increasingly becoming the focus of experimental and computational studies [69,70,71,72]. To date, specific interactions of GPCRs with cholesterol, ganglioside lipids, polyunsaturated lipids, and phosphatidylinositol lipids have been observed in an environment markedly distinct from the surrounding membrane.

### 8.1. Cholesterol

The majority of the collected data focus on cholesterol, recognized as an essential membrane component. In addition to its pivotal roles in influencing the membrane dimensions, organization, and fluidity, cholesterol is actively involved in interactions with diverse membrane proteins, including GPCRs, and plays a well-established role in signal transduction and receptor oligomerization [73,74,75,76]. Cholesterol’s interaction with GPCRs, such as the beta-2-adrenergic receptor, occurs in an allosteric manner, influencing its conformational distribution [77]. Notably, cholesterol has been shown to potentially stabilize the dimerization interface for various class A GPCRs [78]. An effect of cholesterol on GPCR oligomerization can occur either directly, where cholesterol is involved in stabilizing the dimeric interface depending on the state of the receptor interface, or indirectly, when cholesterol affects the organization of the cell membrane, subsequently influencing oligomerization. Cholesterol has the propensity to associate into cluster-forming membrane nanodomains termed lipid rafts and substantially affect signal transduction [79,80,81,82], and is furthermore likely to regulate the formation of GPCR oligomers [83,84,85].

The direct role in dimerization has been demonstrated, among others, for A2A adenosine receptors, β2-adrenergic receptors, 5-HT1A receptors, and μ-opioid receptors [86,87,88,89]. For dopamine receptors, the role of cholesterol has not been definitively established. The D1 receptor appears to be sensitive to cholesterol, unlike D2, but there are insufficient data to confirm this conclusively. Far-resolved structures of the D1 receptor revealed a significant variation in the number and position of cholesterol, not allowing for a clear conclusion regarding the effect of the receptor state on cholesterol binding [90]. It is worth noting that these resolved structures were primarily determined with agonists. However, it has been confirmed that the cellular membrane localization of the D1 receptors depends on cholesterol [91]. The disruption of cholesterol and, consequently, the lipid rafts, through the depletion and removal of cholesterol using methyl-β-cyclodextrin, has resulted in the enrichment of the D1 receptor population within certain membrane regions, leading to an increase in their homo-oligomerization. A similar response was observed when cellular sphingolipid levels were reduced by Fumonisin B1, another important component of lipid rafts [91]. In contrast, the D2 receptor remained unaffected by cyclodextrin-induced cholesterol depletion [92]. In summary, D1 receptors localize in raft domains, although this localization is observed only in certain receptor conformations [91,93,94,95]. The role of lipid rafts in the regulation of D2 receptors, which are diffusely distributed across membrane domains, remains poorly understood [92,93,96]. Therefore, the involvement of lipid domains in the regulation of both D1- and D2-class dopaminergic receptors, especially their heterooligomers, is complex and requires further investigation.

### 8.2. Polyunsaturated Fatty Acids

Polyunsaturated fatty acids, essential components of brain cell membranes, serve as significant modulators of the spatial regulation of membrane physical properties, including fluidity, flexibility, and curvature, directly influencing the activity of GPCRs [97,98,99,100]. They not only affect the binding of ligands to receptors, but also modulate their oligomerization, either strengthening or weakening it. Similar to cholesterol, their action may alter general membrane properties, as well as direct interactions with receptor molecules. Polyunsaturated fatty acids facilitate receptor molecules partitioning into ordered membrane domains [101]. This phenomenon has been observed in the homo-oligomerization of A2AR and its hetero-oligomerization with D2R, where an increase in the membrane concentration of docosahexaenoic acid (DHA) leads to the clustering of DHA around both receptors, thereby enhancing protein heteromerization [102]. This assembly is primarily driven by kinetics, resulting from an increase in receptor lateral diffusion and the rate of spontaneous receptor–receptor interactions. Moreover, DHA also enhances D2 receptor ligand binding affinity and strongly influences conformational dynamics of the receptor, notably of the second intracellular loop, affecting β-arrestin coupling [103]. Polyunsaturated fatty acids can also weaken oligomerization. They accumulate at the protein surface through specific interactions with distinct regions of the GPCR, thereby impeding the formation of compact dimers. For example, in the case of the neurotensin receptor NTS1, unsaturated lipids surrounding NTS1 appear to act as a shield, preventing transmembrane segments from engaging in direct protein–protein contacts and inhibiting compact dimer formation [104]. While the number of up-to-date research literature is quite limited, it does underscore the importance of polyunsaturated fatty acids in GPCR function.

### 8.3. Anionic Lipids

GPCRs demonstrate a pronounced preference for anionic lipids over zwitterionic ones. A proposed concept suggests that the type of lipid can alter the packing at protein–protein and/or protein–lipid interfaces, thereby modulating protein activity [105]. In the case of GPCRs, lipid headgroups not only impact the membrane partitioning of receptor molecules, but also play a role in ternary interactions by influencing the coupling between the lipid membrane and receptor–effector proteins. Anionic lipids have been found to enhance the stability of the receptor’s active state and prolong its lifetime, as demonstrated in the presence of phosphatidylglycerol for the adrenergic β2 receptor [106,107]. Moreover, phosphatidylinositol stabilizes the G protein-bound active state of β1, A2A, and NTR1 receptors, while enhancing the selectivity of coupling to G proteins [108]. Additionally, unique interactions have been observed between G proteins and the dopamine D2 receptor involving specific lipid headgroups [109,110]. These examples illustrate that anionic lipids can insert between receptor helices, forming a salt bridge and blocking the receptor in a specific conformation as it awaits interaction with downstream signaling proteins.

### 8.4. G Proteins

Lipid–protein interactions extend well beyond receptors to include most signaling proteins. They are particularly prevalent in G proteins, G protein-coupled effector enzymes, and receptor kinases [111,112,113,114,115,116]. For the signal to propagate, the proper reciprocal localization of G proteins and receptors is required. Post-translational fatty acid modifications are crucial for an accurate membrane localization of G proteins. This process involves three types of acylation: myristylation, palmitoylation, and isoprenylation. All mammalian Gα proteins, except for Gαt and Gαgust, undergo palmitoylation, the post-translational attachment of a 16-carbon palmitate to at least one cysteine in the N-terminal region of the protein [117]. Palmitoylation is a reversible process. After activation of the G protein by the receptor, the palmitoyl residue detaches, causing depletion of the membrane fraction [118,119]. Myristylation is a process that occurs only in Gαi proteins, where a 14-carbon saturated fatty acid is added to the N-terminal glycine. Unlike palmitoylation, this process is irreversible [120]. The Gγ subunits are affected by the third type of lipidation, prenylation, which involves the irreversible attachment of a 15-carbon farnesyl or 20-carbon geranylgeranyl to the C-terminal cysteine [121]. The main purpose of lipidations is to anchor proteins to the cytoplasmic side of the cell membrane. Studies have shown that in some cases, covalent lipid modifications are not sufficient and an additional signal determining membrane localization is necessary: the presence of a polybasic region at the N-terminus of the G protein [111,122]. Furthermore, electrostatic interactions between positively charged amino acids and the negatively charged surface of the cell membrane provide additional support for membrane localization.

It is generally agreed that G protein signaling through GPCRs is dependent on their membrane localization. Studies have shown that the Gαi1-3 subunits, which have long been thought to be identical, have distinct membrane localizations that influence signal transduction, which is reflected in the changes in intracellular cAMP concentration [113]. Preventing the attachment of lipid anchors to G proteins results in impaired signal transduction. For instance, when the N-terminal glycine or its adjacent cysteine in the Gαs subunit were mutated, membrane localization decreased. This effect was reversed when the remaining components of the heterocomplex, the Gβ1γ2 proteins, were present [122,123]. Analogous studies conducted on the Gαi3 subunit revealed that the subunit lacking lipid anchors exhibits weaker membrane localization. It can only be retrieved in the presence of specific βγ dimers [123,124].

### 8.5. Lipidation of GPCRs

Considering the influence of lipids on receptor functions, dimerization, and their spatial-temporal interaction with G proteins, it can be concluded that lipidation affects signal transduction through dimers. In addition, GPCRs can undergo palmitoylation, but the effects of this modification are unpredictable and vary depending on the specific GPCR. Palmitoylation impacts all aspects of GPCR signaling and may influence interactions with certain G proteins [125]. This has been demonstrated for opioid receptors, where the interaction of cholesterol with the palmitoylated μ-opioid receptor facilitated homodimerization and conjugation with the Gai2 protein [126]. Furthermore, more research suggests that the lipidation of some GPCRs, particularly palmitoylation, facilitates receptor compartmentalization in lipid rafts and dimerization [127].

### 8.6. Active Role of Lipids in GPCR–Ligand Interactions

Discussing the roles of lipids and membranes in the functioning of GPCRs, it is crucial to highlight another significant aspect: the influence of lipids and membrane surfaces on the diffusion and dynamics of neurotransmitters and drugs that target membrane proteins (reviewed in [128]). The potential involvement of membrane sorting should be considered for small molecules or ligands. If a drug is designed to reach a cell membrane-embedded or an extracellular ligand-binding site, its binding rate could be augmented through the sorting process. This is something that should be kept in mind when designing and studying bifunctional ligands or the ones targeted to GPCR heterodimers. As GPCRs, but also heterodimers, are increasingly being identified in intracellular compartments [129], the nature of the given ligand and the role of membrane lipids in the signal transduction processes under investigation should be taken into account even more so. Within the context of lipid and membrane influences on GPCR dynamics, Table 1 serves as a comprehensive summary, outlining the various factors that influence GPCR function, with a specific focus on cholesterol, lipids, and G proteins.

## 9. Conclusions

The above comment signals new aspects to be taken into account in the search for new therapeutics within the ligands of GPCR heterodimers. Many of even the most sophisticated in vitro models do not fully reflect the in vivo situation, particularly in the central nervous system, where even the exact level of expression of a given receptor pair on the same cell is difficult to determine, let alone the exact percentage of these receptors forming heterodimers.

Almost every publication on the heterodimerization of a given pair of GPCRs mentions how important a target this will be for potential drugs. Of course, research into the various aspects of the heterodimerization process and the function of GPCRs is extremely important and of great relevance to basic science, but one must be aware that we are far from being able to make full use of the results of this research in clinical settings.

In the search for potential drugs among heterodimer ligands, more attention should be paid to the endogenous environment under which these heterodimers actually function.

In some cases, it has been shown that the heterodimer in question was indeed important in a given pathological situation, whereby either its density is increased, such as D1–D2 in depression, the reciprocal allosteric modulation of protomers is important (as in case of adenosine A2A and dopamine D2 receptors), or the specific location of the heterodimer would allow for the selective modulation of nociceptive stimuli (as in the case of the opioid system). However, despite this knowledge gained through a significant amount of research, there have been no spectacular successes, apart from the adenosine A2A receptor antagonist, introduced into clinical settings.

Perhaps, then, more attention in basic research should be paid to the environment in which heterodimers function: the G-protein subunits which are available in a given cell population (neuronal or non-neuronal), and the lipids which form the membrane of these cells, whether ligands thought to act at the level of extracellular receptor domains, with an affinity for membrane lipids. This may not only be relevant for the activation of receptors (or heterodimers) located in the cell membrane, but may also affect the action of the receptors located in different compartments inside the cell.

It should always be borne in mind that GPCRs, although the most important in the entire signal transduction system and apparently the easiest ligand targets, do not function in isolation from either protein or lipid partners.

## Figures and Tables

**Figure 1 ijms-25-03089-f001:**
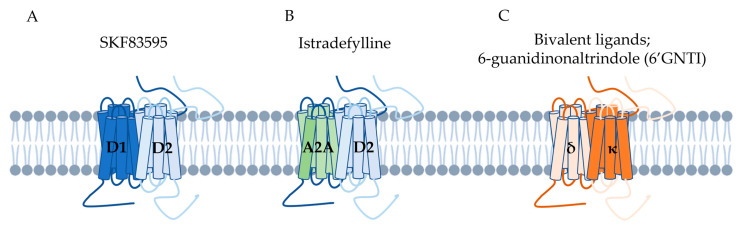
Examples of the compounds described above acting through GCPR dimers: ligand SKF83595 affects the dimerization of dopamine D1 and D2 receptors; Istradefylline, (commercialized as Nourianz^®^) is a adenosine A2A and dopamine D2 receptor heterodimer-selective ligand; while 6-guanidinonaltrindole (6′GNTI) is a delta (δ) and kappa (κ) receptor heteromers agonist.

**Table 1 ijms-25-03089-t001:** Summary of the factors influencing GPCR function: cholesterol, lipids, and G proteins.

Lipid Component	Role in GPCR Function	Citations
Cholesterol	Plays a role in signal transduction and receptor oligomerization Stabilizes dimerization interface for various class A GPCRs Can affect GPCR oligomerization directly or indirectly through membrane organization Associates into lipid rafts, affecting signal transduction and regulating GPCR oligomers Directly involved in the dimerization of A2A adenosine receptors, β2-adrenergic receptors, 5-HT1A receptors, and μ-opioid receptors D1 dopamine receptor is sensitive to cholesterol, affecting localization, and homodimerization, while the D2 receptor remains unaffected	[73,74,75,76,77,78,79,80,81,82,83,84,85,86,87,88,89,90,91,92,93,94,95,96]
Polyunsaturated Fatty Acids	Modulators of membrane physical properties, influencing GPCR activity Affect ligand binding and modulate oligomerization Facilitate partitioning of receptor molecules into ordered membrane domains Influence homo-oligomerization and hetero-oligomerization of receptors like A2AR and D2R Enhance D2 receptor ligand binding affinity and conformational dynamics Can weaken oligomerization by impeding the formation of compact dimers, as observed in the neurotensin receptor NTS1	[97,98,99,100,101,102,103,104]
Anionic Lipids	GPCRs show a preference for anionic lipids over zwitterionic ones Impact membrane partitioning of receptor molecules and play a role in ternary interactions Enhance stability of the receptor’s active state and prolong its lifetime Unique interactions observed between G proteins and the dopamine D2 receptor involving specific lipid headgroups	[105,106,107,108,109,110]
G Proteins	Lipid–protein interactions prevalent in G proteins, G protein-coupled effectors, and receptor kinases Post-translational fatty acid modifications crucial for accurate membrane localization Lipidations anchor proteins to the cytoplasmic side of the cell membrane G protein signaling through GPCRs is dependent on membrane localization Distinct membrane localizations of Gαi1-3; subunits influence signal transduction	[111,112,113,114,115,116,117,118,119,120,121,122,123,124]
Lipidation of GPCRs	Lipidation affects signal transduction through dimers GPCRs can undergo palmitoylation, impacting various aspects of signaling and interactions with G proteins Palmitoylation facilitates receptor compartmentalization in lipid rafts and dimerization, as seen in opioid receptors	[125,126,127]
Active Role of Lipids in GPCR–Ligand Interaction	Lipids and membrane surfaces influence the diffusion and dynamics of neurotransmitters and drugs targeting membrane proteins Membrane sorting may augment the binding rate of ligands designed to reach cell membrane-embedded or extracellular ligand-binding sites	[128,129]

## Data Availability

Not applicable.

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
