# Peer review of "G Protein-Coupled Receptor Dimerization—What Next?"

_ijms, 2024, doi:10.3390/ijms25063089_

Round 1

Reviewer 1 Report

Comments and Suggestions for Authors

The authors have comprehensively reviewed the current studies related to GPCR dimerizations and its applications or implications for treatments of different diseases. They provided detailed examples of heterodimerization of GPCRs in the nervous system and tried to demonstrate that targeting heterodimers located in specific neurons may lead to minimum unwanted side effects. A good example about this is for dopamine D1 and D2 receptor heterodimers. The authors further provided examples and thoughts about other heterodimers in addiction and other psychiatric diseases. Finally they described the impact of plasma membrane on GPCR functioning. Taken together the authors have done a good job in summarizing the importance of GPCR heterodimerization. The following lists out some suggestions that can improve the readability of the manuscript.

The authors need to shorten sentences and enable it to be concise enough. It is important to ensure the writing without language issues, as some sentences cannot be understood.

Whenever explaining well studied GPCR heterodimers, it is suggested to visualize the results along the descriptions in the main text. Currently, the paper lacks of figures to display the structures of explained heterodimers and how they interact or are affected by other molecules.

It seems there is no successful story in treating specific diseases by targeting herterodimers in the manuscript. The authors should discuss the difficulties related to this and propose future directions to resolve the problems.

Comments on the Quality of English Language

The English is acceptable.

Author Response

Reviewer 1

Comments and Suggestions for Authors

The authors have comprehensively reviewed the current studies related to GPCR dimerizations and its applications or implications for treatments of different diseases. They provided detailed examples of heterodimerization of GPCRs in the nervous system and tried to demonstrate that targeting heterodimers located in specific neurons may lead to minimum unwanted side effects. A good example about this is for dopamine D1 and D2 receptor heterodimers. The authors further provided examples and thoughts about other heterodimers in addiction and other psychiatric diseases. Finally they described the impact of plasma membrane on GPCR functioning. Taken together the authors have done a good job in summarizing the importance of GPCR heterodimerization. The following lists out some suggestions that can improve the readability of the manuscript.

Thank you very much.

The authors need to shorten sentences and enable it to be concise enough. It is important to ensure the writing without language issues, as some sentences cannot be understood.

Altered selected long sentences. Changes have been marked in yellow.

Whenever explaining well studied GPCR heterodimers, it is suggested to visualize the results along the descriptions in the main text. Currently, the paper lacks of figures to display the structures of explained heterodimers and how they interact or are affected by other molecules.

A figure summarizing the paragraphs on GPCR heterodimerization has been added. 

It seems there is no successful story in treating specific diseases by targeting herterodimers in the manuscript. The authors should discuss the difficulties related to this and propose future directions to resolve the problems.

It appears that the issue lies with an overly complex research framework, which we attempted to describe in the paragraphs concerning the influence of the plasma membrane on GPCR functioning. 

Comments on the Quality of English Language

The English is acceptable.

Thank you very much.

Reviewer 2 Report

Comments and Suggestions for Authors

The manuscript collects results demonstrating the involvement of GPCR heterodimer formation into progression of brain disorders, and how drugs targeting only GPCR heterodimers rather than their monomeric components can be more effective, selective, and cause less side effects. In addition, authors consider how the plasma membrane can affect GPCR heterodimer formation. Overall, the manuscript is clearly written and provides interesting information for specialists studying brain disorders and specialists working on the development of GPCR-targeting drugs.

1. My main concern is a slight inconsistency between the abstract and the main text. The abstract states that the review considers only "intricate interplay between lipids and GPCRs as a potential key factor in understanding the complexity of cell signaling". The interplay between lipids and GPCRs is considered only in Part 8, and systematization of this information is provided in Table 1. 

On the other hand, Parts 3-7 of the manuscript cover information on GPCR heterodimerization involvement in brain disorders. This is not reflected in the abstract of the manuscript. Moreover, a summarization of information provided in Parts 3-7 in a table similar to Table 1 would strengthen the manuscript. 

2. Lines 50-55:  "First evidence that the interaction and mutual regulation between GPCRs, and the second one – concerning opioid receptors kappa and delta, take place came in 70s from biochemical studies indicating the negative cooperativity between beta-adrenergic receptors [6], and from behavioral studies concerning adenosine and dopamine receptors [7]”. 

The sentence is a bit confusing and needs to be rewritten. It would be more logical to put the phrase "and the second one – concerning opioid receptors kappa and delta" in an additional sentence or removed.

3. Line 191: replace dot with comma

Author Response

Reviewer 2

The manuscript collects results demonstrating the involvement of GPCR heterodimer formation into progression of brain disorders, and how drugs targeting only GPCR heterodimers rather than their monomeric components can be more effective, selective, and cause less side effects. In addition, authors consider how the plasma membrane can affect GPCR heterodimer formation. Overall, the manuscript is clearly written and provides interesting information for specialists studying brain disorders and specialists working on the development of GPCR-targeting drugs.

Thank you very much.

My main concern is a slight inconsistency between the abstract and the main text. The abstract states that the review considers only "intricate interplay between lipids and GPCRs as a potential key factor in understanding the complexity of cell signaling". The interplay between lipids and GPCRs is considered only in Part 8, and systematization of this information is provided in Table 1.

The abstract has been modified.

On the other hand, Parts 3-7 of the manuscript cover information on GPCR heterodimerization involvement in brain disorders. This is not reflected in the abstract of the manuscript. Moreover, a summarization of information provided in Parts 3-7 in a table similar to Table 1 would strengthen the manuscript.

A figure summarizing the paragraphs on GPCR heterodimerization has been added. 

Lines 50-55:  "First evidence that the interaction and mutual regulation between GPCRs, and the second one – concerning opioid receptors kappa and delta, take place came in 70s from biochemical studies indicating the negative cooperativity between beta-adrenergic receptors [6], and from behavioral studies concerning adenosine and dopamine receptors [7]”.

The sentence is a bit confusing and needs to be rewritten. It would be more logical to put the phrase "and the second one – concerning opioid receptors kappa and delta" in an additional sentence or removed.

It has been rewritten. All changes in the manuscript text have been highlighted in yellow.

Line 191: replace dot with comma

It has been done.

Round 2

Reviewer 1 Report

Comments and Suggestions for Authors

Whenever possible, please shorten sentences in the manuscript for better understanding by readers.

Comments on the Quality of English Language

The English is fine but would be improved if possible.